# Phytochemical Analysis and Antioxidant Effects of *Prunella vulgaris* in Experimental Acute Inflammation

**DOI:** 10.3390/ijms25094843

**Published:** 2024-04-29

**Authors:** Camelia-Manuela Mîrza, Tudor-Valentin Mîrza, Antonia Cristina Maria Odagiu, Ana Uifălean, Anca Elena But, Alina Elena Pârvu, Adriana-Elena Bulboacă

**Affiliations:** 1Department of Morpho-Functional Sciences, Discipline of Pathophysiology, Iuliu Haţieganu University of Medicine and Pharmacy, 400012 Cluj-Napoca, Romania; camelia.mirza@umfcluj.ro (C.-M.M.); uifaleanana@gmail.com (A.U.); ancaelenabut@gmail.com (A.E.B.); parvualinaelena@yahoo.com (A.E.P.); adriana.bulboaca@umfcluj.ro (A.-E.B.); 2Department of Epidemiology of Communicable Diseases, National Institute of Public Health—Regional Centre of Public Health, 400376 Cluj-Napoca, Romania; 3Department of Environmental Engineering and Protection, Faculty of Agriculture, University of Agricultural Sciences and Veterinary Medicine, 400372 Cluj-Napoca, Romania

**Keywords:** *Prunella vulgaris*, experimental inflammation, oxidative stress, antioxidant effect, redox balance, phytochemical analysis, phenolic compounds, flavonoids, total phenolic content, HPLC-DAD-ESI, DPPH assay

## Abstract

*Prunella vulgaris* (PV) is one of the most commonly used nutraceuticals as it has been proven to have anti-inflammatory and antioxidant properties. The aim of this study was to evaluate the phytochemical composition of PV and its in vivo antioxidant properties. A phytochemical analysis measuring the total phenolic content (TPC), the identification of phenolic compounds by HPLC-DAD-ESI, and the evaluation of the in vitro antioxidant activity by the DPPH assay of the extract were performed. The antioxidant effects on inflammation induced by turpentine oil were experimentally tested in rats. Seven groups with six animals each were used: a control group, the experimental inflammation treatment group, the experimental inflammation and diclofenac sodium (DS) treatment group, and four groups with their inflammation treated using different dilutions of the extract. Serum redox balance was assessed based on total oxidative status (TOS), nitric oxide (NO), malondialdehyde (MDA), total antioxidant capacity (TAC), total thiols, and an oxidative stress index (OSI) contents. The TPC was 0.28 mg gallic acid equivalents (GAE)/mL extract, while specific representatives were represented by caffeic acid, *p*-coumaric acid, dihydroxybenzoic acid, gentisic acid, protocatechuic acid, rosmarinic acid, vanillic acid, apigenin–glucuronide, hesperidin, kaempferol–glucuronide. The highest amount (370.45 μg/mL) was reported for hesperidin, which is a phenolic compound belonging to the flavanone subclass. The antioxidant activity of the extracts, determined using the DPPH assay, was 27.52 mmol Trolox/mL extract. The PV treatment reduced the oxidative stress by lowering the TOS, OSI, NO, and MDA and by increasing the TAC and thiols. In acute inflammation, treatment with the PV extract reduced oxidative stress, with lower concentrations being more efficient and having a better effect than DS.

## 1. Introduction

Oxidative stress is a phenomenon caused by an imbalance between the production and accumulation of ROS (reactive oxygen species) in cells and tissues. ROS, resulting from various physiological processes in the body, are unstable molecules and exert harmful effects when imbalanced with antioxidant activity. As ROS molecules can cause changes in cellular structures, such as membranes, proteins, lipids, lipoproteins, and deoxyribonucleic acid [1], the measurement of specific biological biomarkers is important in the evaluation of redox state, which is useful for treatments targeting oxidative stress-associated with various disorders, and in preventing damage and oxidative stress-associated disorders [2]. One of the most important oxidative stress biomarkers is malondialdehyde (MDA), which is used for assessing lipid peroxidation [2]. The effects of free radicals and oxidative stress are counteracted by endogenous enzymatic—superoxide dismutase (SOD), catalase (CAT), and glutathione peroxidase (GPx)—and nonenzymatic—lipoic acid, glutathione, L-arginine, and coenzyme Q10—antioxidant molecules. Furthermore, numerous exogenous molecules of vegetal or animal origin have antioxidant activities, mainly introduced into the body via diet [3,4].

Nitro-oxidative stress results from the excessive production of reactive nitrogen species (RNS), comprising nitric oxide (NO^•^) and peroxynitrite anion (ONOO^−^). Therefore, the suppression of RNS production is correlated with anti-inflammatory activity, and NO is a sensitive biomarker for anti-inflammatory effects [5].

Inflammation and oxidative stress are interrelated because inflammation induces oxidative injury, and oxidative stress triggers inflammation. Since inflammation is associated with oxidative stress, the evaluation of the biomarkers of both processes should be considered in the diagnosis of inflammatory diseases [6].

A World Health Organization (WHO) analysis found that about 80% of the world population uses phytotherapy in primary health care, and most of these phytotherapies use plant extracts and their active components. In particular, nutraceuticals are widely used as alternative or adjuvant therapies due to their beneficial effects and fewer side effects [7].

*Prunella* is a perennial herbaceous plant genus from the Labiatae family. About 15 species of *Prunella* exist worldwide, and the most-studied *Prunella* species is *Prunella vulgaris* L. [8]. PV, commonly known as “self-heal”, is widely distributed in East Asia, Europe, and North America and has been used as a traditional medicine for thousands of years. In Europe, the Celts considered it to be one of the six most useful medicinal plants [9]. It is rich in chemical components and exerts a variety of pharmacological activities such as hypolipidemic, antihypertensive, hypoglycemic, antitumor, liver protection, antibacterial, antiviral, anti-inflammatory, antioxidant, and immune system response regulation activities [7,10,11,12,13,14,15]. Being an astringent plant, it has been used since ancient times for the treatment of minor injuries, wounds, burns, contusions, and tissue repair, administered topically or internally. The infusion of self-heal has also been used to treat inflammation and lesions of the mouth and throat regardless of their cause [16].

Over the years, phytochemical studies on PV have isolated and identified more than 250 compounds, most of which are one of seven types of chemical components: triterpenoids, sterols, flavonoids, phenylpropanoids, essential oils, organic acids, and polysaccharides [17]. Some of the most important compounds, proven to have antioxidant and anti-inflammatory properties, are represented by the phenolic compounds. These compounds contain groups of bioactive molecules: flavonoids, stilbenes, lignans, and phenolic acids that have been proven to have effective antioxidant activity due to their chemical structure [18].

The aim of this study was to perform a phytochemical analysis of the PV ethanol extract by high-performance liquid chromatography (HPLC) and to study its in vitro and in vivo antioxidant activity. The in vitro antioxidant activity was determined using the 2,2-diphenyl-1-picrylhydrazyl (DPPH) assay. The in vivo antioxidant effects were assessed using an experimental model of turpentine-oil-induced inflammation in rats. The potential effectiveness of PV could be useful as a single or adjuvant therapy in oxidative/nitrosative-stress-related disorders.

## 2. Results

### 2.1. Phytochemical Analysis

For the alcoholic extract of PV that was used experimentally, the total phenolic content (TPC) was 0.28 mg GAE/mL extract. The phenolic compounds identified in the PV extract can be framed within three classes and four subclasses. The highest amount (370.45 μg/mL) was reported for hesperidin, which is a phenolic compound belonging to the flavanone subclass (Table 1). 

Seven out of ten phenolic compounds are hydroxybenzoic acids, with rosmarinic acid as the major compound (116.08 μg/mL). The other two compounds are the flavonol-representative kaempferol–glucuronide (53.42 μg/mL) and the flavone representative apigenin–glucuronide (30.96 μg/mL). Gentisic acid (8.88 μg/mL) is the phenolic compound identified in the smallest amounts (Table 1 and Figure 1).

### 2.2. In Vitro Antioxidant Activity

The in vitro antioxidant activity was evaluated using the DPPH method and showed an equivalent factor (F) of 27.52 mmol Trolox/mL extract with an inhibition percentage (I%) of 41. The equivalent factor (F) is a measure of the antioxidant activity of the concerned compounds. The inhibition percentage (I%) provides a quantitative measure of the ability of the sample to neutralize or scavenge free radicals. 

### 2.3. In Vivo Redox Status Parameter Assessment

The assessed serum redox status parameters were malondialdehyde (MDA), total oxidant status (TOS), nitric oxide (NO), total thiol content, total antioxidant capacity (TAC), and oxidative stress index (OSI).

Compared with the control group (CTRL), the MDA values (μmol/L) were significantly higher in the DICLO and PV100 groups. Compared with the inflammation group (INF), MDA had significantly lower values in all treated groups. The PV effect was dose-dependent, with a higher dilution having the best inhibitory activity. The PV25 and PV10 treatments had better inhibitory effects on MDA than DS (Table 2).

The TOS values (μmol H_2_O_2_ Eq/L) were significantly higher in all groups that received treatment compared with the CTRL group. Compared with the INF group, significantly lower values were observed in the DICLO, PV100, and PV25 groups. In addition, compared with the DICLO group, the TOS values were higher in all groups treated with the PV extract, with the differences being significant for the PV100, PV50, and PV10 groups (Table 2).

The NO values (μmol/L) did not differ significantly for any of the treated groups compared with the CTRL group. Compared with the INF group, lower values were observed in all treated groups, with the differences being significant. Among the groups that received treatment, the best inhibitory activity was found in the groups treated with PV, and the effect was dose-dependent, with a higher dilution having the better inhibitory activity (Table 2). 

Compared with the CTRL group, total thiols values (μmol/L) were lower in all the other groups. Compared with the INF group, total thiols increased very significantly in the PV50 and PV10 groups and increased slightly in the PV25 group. Compared with the DICLO group, except for the PV100 group, the PV extract dilutions increased total thiols, with the highest values being observed in the PV50 and PV10 groups (Table 2).

The TAC values (mmol Trolox Eq/L) did not differ significantly for any treated group compared with the CTRL group. Compared with the INF group, the TAC values were significantly higher in the DICLO, PV100, and PV50 groups. All groups treated with the PV extract, regardless of concentration, had higher TAC values compared with the DICLO group, with the differences being significant in the PV50 group (Table 2).

The OSI had significantly higher values in all groups compared with the CTRL group. Compared with the INF group, the OSI had lower values in all groups that received treatment, with the differences being significant in the PV100, PV25, and PV10 groups. The OSI values were lower in the DICLO group compared with the groups that received the self-heal extracts, with the differences being significant for the PV100, PV50, and PV10 groups (Table 2).

### 2.4. Statistical Correlation Analysis between the Studied Redox Status Parameters

In the seven studied groups, several very good or good correlations were observed between the redox status parameters, with some of them being significant (Table 3).

NO and TOS were positively correlated in the CTRL, DICLO, PV50, and PV10 animals. NO and TAC were also positively correlated in the INF, PV100, PV50, PV25, and PV10 animals. The only good correlation between MDA and TOS was observed in the PV10 group. Thiols and TAC were positively correlated in the CTRL and INF animals. Additionally, TOS and thiols were positively correlated in the CTRL and PV50 animals but negatively correlated in the PV25 ones. MDA and TAC were positively correlated in the INF animals but negatively correlated in the CTRL and PV25 animals (Table 3).

## 3. Discussion

Higher amounts of rosmarinic acid are identified in our trial compared with those reported by Sárosi et al. in a study on the influence of PV geographical origin and climatic Conditions on TPC and antioxidant capacity [19]. Sahin et al. identified relatively similar amounts of caffeic acid and rosmarinic acid in several PV species, unlike our findings, which show high differences between the amounts of the above-mentioned compounds [20,21], but results reported by Golembiovska which show higher amounts of rosmarinic acid compared with caffeic acid in PV, are consistent with our findings [22]. The presence of protocatechuic acid in PV extracts was reported in the literature by Sahin et al. but in much lower concentrations compared with rosmarinic acid and caffeic acid [21].

The phytochemical analysis of the PV ethanol extract determined which of the compounds had antioxidant activities and confirmed that the extract effectively reduced turpentine oil inflammation-induced oxidative stress. Lower concentrations were more efficient compared with higher concentrations of PV, and these effects were better than those of DS (Table 2). Other studies have also demonstrated that the components of the PV extract have antioxidative activities. For example, dihydroxybenzoic acid inhibits increases in adhesion molecules, inflammatory cytokines, and MAPK levels, as well as NO production, and contributes to the prevention of rises in ROS generation [23]. Gentisic acid (2,5-dihydroxybenzoic acid), a component of the PV extract, is extensively utilized in the pharmaceutical sector and has been documented to exhibit anti-inflammatory, antioxidant anticarcinogenic, and antimutagenic activities [24,25]. Protocatechuic acid is a derivative of benzoic acid, which is water-soluble and is the principal metabolite of polyphenols from herbs, vegetables, and fruits. It has been reported to have anti-inflammatory, antibacterial, antiviral, and more recently confirmed, antioxidant properties [26,27]. Laboratory studies conducted on rats have shown the positive effect of vanillic acid on oxidative stress and antioxidant enzymes [28,29], while other studies developed according to murine models have shown the role of vanillic acid in decreasing the production of pro-inflammatory cytokines, thus emphasizing its anti-inflammatory and analgesic effects [29,30]. Paciello et al. demonstrated the role of caffeic acid in reducing oxidative stress and inflammatory pathways by decreasing NF-kB (nuclear factor kappa-light-chain-enhancer of activated B cells) and IL-1βb (interleukin-1-beta) expression in the cochlea of Wistar rats with hearing loss induced by noise [31]. Testing the role of *p*-coumaric acid in neurodegenerative diseases such as Alzheimer’s disease, Kan et al. emphasized its anti-amyloidogenic qualities and natural antioxidative characteristics, which may prove beneficial in formulating a neuroprotective agent [32]. Studies concerning hesperidin, which is a flavone mostly identified in citrus fruits, showed its effect on lowering the levels of antioxidant and/or anti-inflammatory parameters such as C reactive protein (CRP); interleukins 4 and 6 (IL-4 and IL-6); and malondialdehyde (MDA), a lipid peroxidation marker [33]. Rosmarinic acid is a monomeric constituent isolated from more than 160 plants. Studies demonstrated its anti-inflammatory action by inhibiting IL-6 production and even cartilage extracellular matrix degradation in osteoarthritis [34,35]. Lin et al. suggested a strong antioxidant activity of rosmarinic acid on the mitogen-activated protein kinase (MAPK) pathways and in modulating insulin/insulin-like growth factor signaling (IIS), with a high survival rate of mice exposed to thermal and oxidative stress [36]. Kaempferol–glucuronide is a primary metabolite of kaempferol, which is a bioactive glycoside. Park et al. showed the antioxidant effect of kaempferol–glucuronide, demonstrating its scavenging capacity for 1,1-diphenyl-2-picrylhydrazyl radicals and inhibition activity on xanthine/xanthine oxidase [37]. Apigenin–glucuronide is a flavonoid identified as having significant contributions to the antioxidant potential of extracts from wild thyme [38,39]. 

It is known that in PV, the most important components with medical effects are accumulated mainly in the dry fruit spike but can also be found in the rest of the herb [8]. The ethanol extract we used was prepared from a whole plant product. Because in Europe, PV is approved as a nutritional supplement [40], we considered a detailed analysis of the antioxidant activity in relation to the chemical composition necessary.

Initially, we performed a phytochemical analysis of the extract (Table 1). Studies that compared the effect of various solvents on the TPC of the whole-plant PV extract found that ethanol was the best solvent for phenol extraction. Tosun et al. identified 336.45 mg GAE/g TPC in the whole-plant ethanol extract [41]. Differences may have been observed because the chemical composition of plant extracts varies with the extraction method, plant organ and overall growth phases, harvesting season, and geographical area and climate [40].

Many phytochemical studies of PV have measured the most important chemical compounds, such as polysaccharides, phenolic acids, triterpenoids, flavonoids, and tannins because they have important pharmacological effects [10]. Our HPLC-DAD-ESI analysis found that PV contains 10 major phenolic compounds. Concerning flavonoids, we found that hesperidin showed the highest concentration, while smaller amounts of kaempferol and apigenin were also present (Table 1). These flavonoids are important for our study because they have a wide range of pharmacological activities, such as antioxidative, free radical scavenging, and anti-inflammatory activities [42]. 

Furthermore, our HPLC-DAD-ESI analysis showed a high content of rosmarinic acid and smaller concentrations of caffeic acid and p-coumaric acid (Table 1). Similarly, the data obtained by other authors indicated that rosmarinic acid and caffeic acid are the main components in all parts of PV, with rosmarinic acid having a higher concentration [17]. However, unlike our results, where 10 major phenolic compounds were identified in PV plants that originated in Croatia, in an LC/MS study performed in China by Feng et al. [14], four major phenolic compounds were identified in a Chinese variety of PV: rosmarinic acid, caffeic acid, rutin, and quercetin. These compounds are specific to PV varieties cultivated in Asian countries such as China, Japan, and Korea [43]. Using an HPLC-ECD analysis, Jirovský et al. [44] identified 35.7 μg/mg and 0.816 μg/mg of rosmarinic acid and caffeic acid, respectively, in the aerial parts of PV cultivated in the Czech Republic. Additionally, an HPLC analysis of PV plants cultivated in the Czech Republic conducted by Psotová et al. [11] led to a similar finding to that in our study: a high content of rosmarinic acid. Other analyses isolated three coumarin compounds. These findings are relevant due to their previously proven pharmacological effects, such as their antioxidant, anti-inflammatory, and immune response regulation effects [42]. Moreover, a PV 60% ethanol extract showed high in vitro antioxidant activity when removing free radicals [17], verified based on the ABTS, DPPH, and FRAP methods. In concordance with that study, our PV ethanol extract showed high in vitro antioxidant activity in the DPPH assay. 

The rich chemical composition and the in vitro antioxidant activity of the PV extract thus create a need to study more aspects of the in vivo antioxidant effect despite several studies having already analyzed the effects of different PV extracts on inflammation and found that the extract considerably reduces inflammation scores as well as the serum levels of some pro-inflammatory cytokines [17,45]. Because the anti-inflammatory activity of antioxidants is not a novel idea, in our in vivo study of the antioxidant activities of the PV ethanol extract, we compared the results obtained by administering the self-heal extract in different concentrations, with those obtained by administering DS, a classic anti-inflammatory agent (Table 2). 

Another study evaluated the prophylactic activity of the PV extract [46]. In the present study, we used the PV extract for in vivo therapy after inducing experimental inflammation. Currently, most of the biomarkers used to assess redox status are the end-products of oxidative stress. First, we evaluated the oxidative stress levels with general tests: TOS, TAC, and OSI. In the INF group, we found a significant level of oxidative stress, expressed by TOS and OSI increasing and TAC decreasing. The results showed that all the PV concentrations reduced TOS, but the effect was smaller than that of DS (Table 2). A TOS reduction by PV was also observed by other researchers [17]. They reported that the antioxidant compounds identified from natural products could exert beneficial effects on inflammation [17]. All concentrations increased TAC, but only the PV50 treatment had a significant effect. PV demonstrated its in vivo effectiveness on TAC, proving its ability to improve antioxidant capacity (Table 2).

The protective effects of the PV extract were also evaluated by calculating the OSI, a ratio between the TOS and the TAC. It increased in the INF group but decreased in PV extract treatments. The inhibitory effect of DS on the OSI was stronger than that of the PV extract (Table 2). 

Nitric oxide production from L-arginine is catalyzed by nitric oxide synthases (NOS), with three different isoforms of NOS enzymes: Two isoforms are expressed constitutively in endothelial cells (eNOSs) and brain tissue (nNOS) [47]. The third NOS isoform, inducible NOS (iNOS), is overexpressed in response to inflammatory stimuli, and relatively large amounts of NO^•^ are produced [48]. The reaction of NO^•^ with the superoxide radical anion (O2^•−^) generates peroxynitrite (ONOO^−^), an RNS that can trigger cell death via oxidation and nitration to form 3-nitrotyrosine residues in proteins. The excessive production of RNS during inflammation causes tissue injury, either directly through DNA damage, lipid peroxidation, protein nitration, and oxidation or indirectly by modulating leukocyte activity [23,43].

In the present study, as expected, in the INF animals, NO was overproduced (Table 2). However, studies analyzing treatments with PV constituents found a decrease in NO, both in vivo and in vitro [3,47,49,50,51]. Since the inhibition of NO• production has been proposed as an anti-inflammatory treatment in inflammation, the lowering effect of our total PV extract on NO synthesis was a positive result. This effect was dose-dependent, with the lower concentration having the best inhibitory activity. Another important finding was that PV extract’s inhibitory effect on NO was better than that of DS. A possible explanation may be that the high concentrations of polyphenols can acquire some pro-oxidant effect, and with each dilution, the antioxidant activity increases [52]. It seems that polyphenols exert two actions: antioxidative actions by ROS scavenging and downregulation of nuclear factor-κB, along with pro-oxidant actions by promoting ROS. However, how these polyphenols produce either pro-oxidant or antioxidant effects remains unclear [53]. Studies have been conducted on polyphenols derived from plants, showing that many dietary polyphenols have both antioxidant and pro-oxidant effects, sometimes dose-related. For example, epigallocatechin-3-gallate (EGCG) from the tea plant *Camellia sinensis* L. Ktze. in large doses has a pro-oxidant effect, while the antioxidant effect is manifested in smaller doses. Another example could be grape seed extracts that have a pro-oxidant activity in vivo, proven to be dependent on the dose and the duration of administration [53,54,55].

Here, inflammation produced a significant increase in MDA values, and low PV extract concentrations succeeded in reducing the effects of inflammation by lowering MDA back to values close to or even at normal values. Similarly, some studies have shown that a 60% ethanol extract of the PV extract can significantly reduce serum MDA and that this extract concentration had the highest content of antioxidant polyphenols, such as caffeic acid, rosmarinic acid, rutin, and quercetin [14,44]. Therefore, MDA reduction may be correlated with the concentration of antioxidant compounds. Furthermore, the inhibitory effects of PV50, PV25, and PV10 on MDA were better than those of DS.

The existence of antioxidant defense systems is an essential condition for the existence and development of cells in the aerobic living environment. Thiols are one of the most important body antioxidants, and the serum concentrations of thiols reflect the systemic redox state. Thiols act as an antioxidant by accepting free radicals unpaired with any electrons. In inflammation, a significantly lower concentration of thiols has been observed, indicating that it is associated with oxidative stress [52,56]. Thus, thiol imbalance could be an early event in oxidative stress [57]. In this study, the treatments with PV50, PV25, and PV10 increased thiols, and for the PV50 and PV10 groups, the effect was better than that of DS (Table 2). These findings suggest that fewer oxidants were generated or that the antioxidant capacity increased. 

Our study showed very good positive correlations between NO and TAC in all groups to which PV was administered, especially in groups with a low concentration of the extract, where they are also significant. In association with a very good negative correlation between thiols and TOS, significant in the PV25 group, our data support the idea of good antioxidant effects of the extract, especially at lower concentrations (Table 3). To our knowledge, this is the first study that has demonstrated the correlative effects of PV on oxidative stress/antioxidant balance.

Our study is also the first to demonstrate the effect of PV compared with a classic non-steroidal anti-inflammatory drug such as diclofenac sodium. The effects of this herbal treatment, in addition to those of DS, could be effective in inflammatory disorders. However, some limitations of this study regarding the anti-inflammatory effects of the PV extract still have to be further demonstrated. Further evaluation of other inflammation biomarkers would be useful to demonstrate the anti-inflammatory activity of PV. Other experimental models could also demonstrate its versatility in various inflammatory conditions.

## 4. Materials and Methods

### 4.1. Chemicals and Reagents

6-hydroxy-2,5,7,8-tetramethylchroman-2-carboxylic acid (Trolox), Na_2_CO_3_, Folin–Ciocâlteu reagent, 2,2-diphenyl-1-picrylhydrazyl (DPPH), 5,5-dithio-bis-(2-nitrobenzoic acid) (DTNB, Ellman’s reagent), Tris-EDTA buffer solution, trichloroacetic acid (TCA), thiobarbituric acid (TBA), methanol, acetonitrile, N-(1-naphthyl)ethylenediamine dihydrochloride (NEDD), sulphanilamide (SULF), vanadium(III) chloride (VCl_3_), sodium nitrite, sodium nitrate, sodium iodide (NaI), reduced glutathione (GSH), hydrochloric acid, phosphoric acid, glacial acetic acid, o-cresosulfonphthalein-3,3-bis (sodium methyl-iminodiacetate) (xylenol orange), horseradish peroxidase, 3,5,3′,5′-tetramethylbenzidine (TMB), ortho-dianisidine dihydrochloride (3-3′-dimethoxybenzidine), ferrous ammonium sulphate, ferric chloride, alchilamine N-N-diethyl-para-phenylenediamine (DEPPD), sodium azide, hydrogen peroxide (H_2_O_2_), tert-butyl hydroperoxide, cumene hydroperoxide, sulfuric acid, glycerol, butylated hydroxytoluene (BHT), 1,1,3-3-tetramethoxypropane (malondialdehyde bis(dimethyl acetal), ethylenediamine tetraacetic acid (EDTA), sorbitol, ferric chloride, ferrous ammonium sulphate, L(+) ascorbic acid (vitamin C), ascorbate oxidase, bilirubin, uric acid, (±)-catechin, 2,4,6-tripyridyl-s-triazine (TPTZ), ethylenediaminetetraacetic acid (EDTA), 2,2′-azino-bis(3-ethylbenz-thiazoline-6-sulfonic acid) (ABTS), hydrogen peroxide, potassium persulphate, and sodium citrate. All chemicals were purchased from Sigma-Aldrich (Burlington, MA, USA) and Merck (Rahway, NJ, USA) and were ultrapure grade, and type I reagent-grade deionized water was used.

### 4.2. Plant Extract

The self-heal organic *Prunella vulgaris* dried herb extract was purchased from Hawaii Pharm LLC, Honolulu, HI, USA (order #65766/15.02.2021; IPN: A-011921-SEPVH-AT; best used by 02/2026). The manufacturer described the product as follows: main ingredients—certified organic self-heal (*Prunella vulgaris*) dried herb (origin: Croatia); other ingredients—vegetable palm glycerin, pharmaceutical-grade alcohol, and water. The extraction ratio of the dry herb material/menstruum in 30% ethanol was 1:3 *w*/*v*. Starting from the 100% PV extract, dilutions of 50%, 25%, and 10% were made using double-distilled water for laboratory use.

### 4.3. Phytochemical Analysis

#### 4.3.1. Total Phenolic Content Analysis

The TPC was determined by measuring the optical density of the PV extract, which, upon complexation with the Folin–Ciocâlteu reagent, absorbs in the visible range (Vis) at 750 nm [58,59]. The total amount of polyphenols was expressed in relation to a calibration curve with gallic acid of different concentrations (1 mg/100 mL, 0.5 mg/100 mL, 0.25 mg/100 mL, 0.125 mg/mL, and 0.0625 mg/mL).

Twenty-four-well microplates with a volume of 3 mL were used. In total, 2.350 mL of distilled water, 0.05 mL of the extract, 0.150 mL of the Folin–Ciocâlteu reagent, and 0.450 mL of sodium carbonate were added to the plate. For the CTRL sample, the 0.05 mL extract was replaced with 0.05 mL methanol. The plates were stored in darkness for 2 h, after which the absorbance was measured using a BioTek multidetection spectrophotometer (BioTek, Winooski, VT, USA) with monochromator-based optics, filter-free, and wide wavelength range (200–999 nm). The TPC was expressed in gallic acid equivalents (GAE)/mL extract. 

#### 4.3.2. Phenolic Compound Identification and Quantification

The analysis was conducted using an Agilent 1200 HPLC (Agilent Technologies, Santa Clara, CA, USA) system equipped with a quaternary pump, a solvent degasser, an auto-sampler, and a UV-Vis photodiode detector (DAD) coupled with an Agilent single quadrupole mass detector (MS) model 6110 (Agilent Technologies, CA, USA). The compounds were separated on a Kinetex XB C18 column, 4.6 × 150 mm, with 5 μm particles (Phenomenex, Torrance, CA, USA), using mobile phases A and B in the gradient below for 30 min at a temperature of 26 °C with a flow rate of 0.5 mL/min. Solvent A: water + 0.1% acetic acid. Solvent B: acetonitrile + 0.1% acetic acid. Gradient (expressed in % B): 0 min, 5% B; 0–2 min, 5% B; 2–18 min, 5%–40% B; 18–20 min, 40%–90% B; 20–24 min, 90% B; 24–25 min, 90%–5% B; 25–30 min, 5% B [60].

The spectral values were recorded in the range of 200–600 nm for all peaks. Chromatograms were recorded at the wavelengths λ = 280 and 340 nm.

For MS, the positive ionization ESI mode was used in the following working conditions: capillary voltage 3000 V; temperature 350 °C; nitrogen flow 7 L/min; nebulization pressure 35 psi; fragmentation voltage 100 eV; *m*/*z* 120–1200, full-scan.

Hydroxybenzoic acids were quantified in gallic acid equivalents (μg/mL), hydroxycinnamic acids in chlorogenic acid equivalents (μg/mL), flavanone in hesperidin equivalents (μg/mL), and flavonol and flavone in rutin equivalents (μg/mL).

A volume of 20 μL of sample was used for quantification. Individual phenolic compounds were identified by comparing their retention times, UV-Vis absorption spectra, and mass spectra with those of available commercial standards and/or with the spectra available in the Phenol-Explorer database [61]. For quantification of phenolic acids, the chromatograms were recorded at 280 nm, while for quantification of flavonoids, the chromatograms were monitored at 340 nm, using DAD detection. Quantification of phenolic acids was performed by external calibration with gallic acid 5–30 µg/mL (for hydroxybenzoic acids) and chlorogenic acid 10–50 µg/mL (for hydroxycinnamic acids), both calibration curves being highly linear (R2 > 0.999). Flavonoids were quantified and expressed as hesperidine equivalents and, respectively, rutin equivalents based on external calibration curves (10–300 µg/mL; R2 > 0.999).

Data acquisition and interpretation were performed using Agilent ChemStation software version E.02.02. 

### 4.4. In Vitro Antioxidant Activity

The antioxidant activity of PV was evaluated in vitro using a modified 2,2′-diphenyl-1-picrylhydrazyl (DPPH) method [62] based on measurements of the antioxidant complexing ability of the DPPH^•^ radical. DPPH^•^ is one of the few stable nitrogen radicals (purple) that fade when discolored by an antioxidant (yellowish). 

The reaction between DPPH and the antioxidants in the extract was monitored using a BioTek spectrophotometer at 515 nm (BioTek, VT, USA). The methanol solution was used as a blank; then, 1.75 mL of DPPH and 250 μL of the sample were used for each determination, with the absorbance being recorded after 30 min. The calibration curve was performed with Trolox using various dilutions (500 μM, 250 μM, and 125 μM to 9.15 μM), and then, the samples’ absorbance was recorded.

The inhibition percentage (I%) was calculated as follows: I% = [(AB − AA):AB] × 100,(1)
where AB = absorbance of the blank solution and AA = absorbance of the standard solution (t = 30 min). 

The antioxidant activity of the evaluated sample, respectively, the equivalent factor—F (mmol Trolox), is reported in terms of 1 mL of the sample extract.

The working protocol for the 24-well plate: 80 µM DPPH was dissolved in 98% methanol. The stock DPPH solution was freshly prepared, sonicated for 15 min, and stored in darkness at room temperature. A 250 µL sample and a 1750 µL DPPH solution were added to the work plate. The control sample contained 250 µL of methanol and 1750 µL of DPPH solution. After 30 min, the absorbance was read at 515 nm. The DPPH solution discolors from violet to yellow in the presence of a hydrogen donor, establishing the degree of inhibition of free radicals (I%). 

### 4.5. In Vivo Antioxidant Activity

#### 4.5.1. Animal Subjects and Experimental Design

The experiment was performed on healthy adult male Wistar rats weighing 220–250 g, obtained from the Animal Facility of Iuliu Haţieganu University of Medicine and Pharmacy Cluj-Napoca. The animals were kept in standard humidity and ventilation conditions at a temperature of 21 ± 2 °C and a 12-h light/dark cycle. They had free access to a standard diet with pellets, including all alimentary elements and water ad libitum. All experimental procedures involving the use of laboratory animals complied with Directive 2010/63/EU and Romanian law 43/2014 concerning the protection of animals used for scientific purposes. The Research Ethics Committee of the Iuliu Haţieganu University of Medicine and Pharmacy Cluj-Napoca approved the experimental protocol (AVZ28/25.11.2021), and the Veterinary Sanitary and Food Safety Authority Cluj approved the experiment (authorization 292/25.02.2022).

Thirty-six animals were randomly divided into six groups (*n* = 6). The following experimental protocol was used to evaluate the anti-inflammatory effect of the PV ethanol extract: INF (positive control), rats with acute experimental inflammation induced by the administration of turpentine [63], and DICLO (anti-inflammatory treatment). Rats given turpentine and receiving sodium DS; CTRL (negative control), healthy rats with a normal diet, kept in standard conditions, with normal values obtained from the database of the Pathophysiology Discipline (the Iuliu Haţieganu University of Pharmacy and Medicine Cluj-Napoca).

In five groups, acute inflammation was induced on day 1 via the administration of turpentine oil, a single intramuscular injection of 0.6 mL/kg b.w. in the left hind paw [59]. Between days 2 and 8, the DICLO group was treated with DS (Terapia, România) as an anti-inflammatory control drug, 10 mg/kg b.w./day p.o., for 7 days, and four groups were treated with different concentrations of the PV extract—PV100 (100% extract), PV50 (50% extract), PV25 (25% extract), and PV10 (10% extract)—administered 1 mL/animal/day p.o. for 7 days. On the 9th day, blood was collected by retro-orbital puncture under general anesthesia with 10% ketamine and 2% xylazine 2:1. Redox status was assessed by measuring serum TOS, TAC, OSI, MDA, SH, and NO. 

#### 4.5.2. Redox Status Evaluation

Redox status parameters were determined in the laboratory of the Pathophysiology Discipline (Iuliu Haţieganu University of Medicine and Pharmacy Cluj-Napoca) using colorimetric methods, with the absorbance (A) of the samples being read on a Jasko V-630 spectrophotometer.

The total antioxidant capacity (TAC) was determined using an automated colorimetric method developed by Erel, in which the characteristic color of ABTS*+ is bleached under the action of the antioxidants present in the serum sample. The reaction is monitored spectrophotometrically. The reaction rate is calibrated with Trolox [64]. The results were expressed in mmol Trolox Eq/L.

The total oxidant status (TOS) was determined using an automatic colorimetric method also developed by Erel, which is based on the oxidation of ferrous ions to ferric ions in the presence of various oxidant species in an acidic medium. The ferric ion makes a colored complex with the xylenol orange reagent, and the absorbance can be measured spectrophotometrically [65]. The results were expressed in μM H_2_O_2_ Eq/L.

The oxidative stress index (OSI) shows the deviation from the normal state of the oxidant/antioxidant balance, with its increase being caused either by the increase in pro-oxidant species or by the decrease in antioxidant protection. It is represented by the ratio between TOS and TAC [66].

The nitric oxide (NO) was determined using the method described by Miranda, Espey, and Wink, based on the simultaneous assessment of nitrate and nitrite concentrations involving a reduction of vanadium (III) nitrate and detection with Griess reagents [67,68]. The results were expressed in μmol/L.

Malondialdehyde, as a peroxidation product of polyunsaturated fatty acids, is an indicator of oxidative stress. The determination of MDA was performed using the thiobarbituric acid method, adapted for a microanalysis, based on the reaction between MDA and TBA, resulting in a TBA-MDA stained complex, which can be quantified spectrophotometrically at 530 nm [69]. The results were expressed in μmol/L.

The thiol assay was based on the color reaction of thiol groups with 5,5-dithio-bis-(2-nitrobenzoic acid) (DTNB or Ellman’s reagent). Briefly, 50 μL of plasma was mixed with 1 mL of 20 mM Tris (0.25M)-EDTA buffer, pH 8.2. Then, 20 μL of the 10 mM DTNB reagent was added, and after 15 min, the absorbance of the yellow solution was read at 412 nm; the DTNB buffer was used as a blank [70]. The results were expressed in μmol/L.

### 4.6. Statistical Analysis

A Shapiro–Wilk test was used to test the normal distribution, and the variance was tested with an F test. In order to summarize the distribution of quantitative variables, the mean ± sample standard deviation (SD) was used. The comparison of two groups in relation to the quantitative characteristics was performed with a Student’s *t*-test or Mann–Whitney U test. The correlation between the parameters of the same group was assessed using Pearson’s (r) or Spearman’s (rho) coefficients in accordance with the Colton scale. The significance thresholds were α = 0.05 (5%), 0.01 (1%), and 0.001 (0.1%).

The statistical data analysis was performed using StatsDirect software v.2.7.2 (StatsDirect Ltd., Wirral, UK).

## 5. Conclusions

The phytochemical evaluation of the PV extract found a significant concentration of antioxidant compounds, and the DPPH test indicated good in vitro antioxidant activity based on DPPH assay results. The ethanol extract we used proved to be rich in rosmarinic acid, as shown by the HPLC assessment. The phenolic compounds identified in the PV extract belong to four subclasses (hydroxybenzoic acids, flavonols, flavanones, and flavones) that have good antioxidant activity. The in vivo treatment of the animals with turpentine-oil-induced inflammation showed that the ethanol PV extract reduced inflammation-dependent oxidative stress in a dose-dependent manner, with lower concentrations (50% or below) having better inhibitory activity. The PV extract’s oxidative stress-reducing effect was better than that of DS, supporting the self-heal extract as a candidate for therapy to reduce the progression of oxidative stress in various diseases. However, the scientific validation of this therapy requires further experimental and clinical studies before introduction into human therapy or as a functional food.

## Figures and Tables

**Figure 1 ijms-25-04843-f001:**
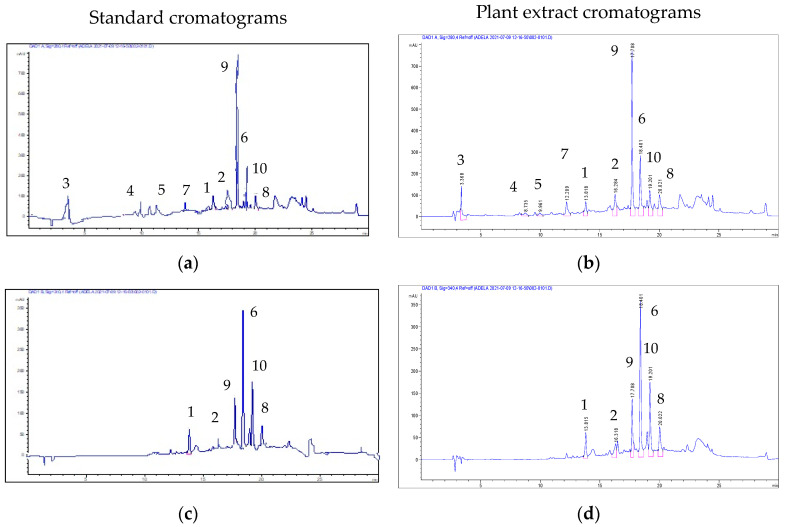
Chromatograms of phenolic compounds identified in PV extract: (**a**) standard chromatogram at λ = 280 nm; (**b**) plant extract chromatogram at λ = 280 nm; (**c**) standard chromatogram at λ = 340 nm; (**d**) plant extract chromatogram at λ = 340 nm. See Table 1 for the identity of the compounds.

**Table 1 ijms-25-04843-t001:** Concentration of phenolic compounds identified in PV extract, μg/mL.

Peak No.	Compound	Class	Subclass	Amount * (μg/mL)
**1**	Caffeic acid	Phenolic acids	Hydroxycinnamic acid	**22.70**
**2**	*p*-Coumaric acid		Hydroxycinnamic acid	**16.99**
**3**	Dihydroxybenzoic acid		Hydroxybenzoic acid	**32.42**
**4**	Gentisic acid		Hydroxybenzoic acid	**8.88**
**5**	Protocatechuic acid		Hydroxybenzoic acid	**7.56**
**6**	Rosmarinic acid		Hydroxycinnamic acid	**116.08**
**7**	Vanillic acid		Hydroxybenzoic acid	**23.23**
**8**	Apigenin–glucuronide		Flavone	**30.96**
**9**	Hesperidin	Flavonoids	Flavanone	**370.45**
**10**	**Kaempferol–glucuronide**		**Flavonol**	**53.42**

* Hydroxybenzoic acids are expressed in gallic acid equivalents (μg/mL), hydroxycinnamic acids in chlorogenic acid equivalents (μg/mL), flavanone in hesperidin equivalents (μg/mL), and flavonol and flavone in rutin equivalents (μg/mL).

**Table 2 ijms-25-04843-t002:** Redox status parameters of the study groups.

Indicator	MDA	TOS	NO	Thiols	TAC	OSI
	(μmol/L)	(μmol H_2_O_2_ Eq/L)	(μmol/L)	(μmol/L)	(mmol Trolox Eq/L)	
**CTRL**	0.884 ± 0.135	4.6163 ± 0.243	27.2322 ± 4.216	331 ± 44.218	1.0879 ± 0.0003	4.2433 ± 0.223
**INF**	1.3607 ^c^ ± 0.063	10.0487 ^b^ ± 2.064	33.4669 ^b^ ± 1.671	259 ^b^ ± 6.693	1.0875 ^a^ ± 0.0002	9.2405 ^b^ ± 1.898
**DICLO**	1.0972 ^bf^ ± 0.078	6.9283 ^cd^ ± 0.652	30.5214 ^d^ ± 2.45	262.6 ^b^ ± 18.216	1.0877 ^d^ ± 0.0001	6.3698 ^c^ ± 0.599
**PV100**	1.118 ^ae^ ± 0.142	7.8931 ^cdg^ ± 0.716	28.9602 ^e^ ± 1.744	257.8 ^b^ ± 14.999	1.0878 ^d^ ± 0.0003	7.2563 ^cdg^ ± 0.658
**PV50**	0.936 ^f^ ± 0.17	9.5098 ^ci^ ± 0.826	29.1664 ^d^ ± 4.196	309.4 ± 61.753	1.0882 ^dg^ ± 0.0003	8.7386 ^ci^ ± 0.758
**PV25**	0.858 ^fh^ ± 0.159	7.7627 ^cd^ ± 0.657	23.5110 ^fi^ ± 2.733	277.4 ^ae^ ± 13.994	1.0878 ± 0.0004	7.1362 ^cd^ ± 0.603
**PV10**	0.7072 ^fi^ ± 0.157	7.8409 ^cdg^ ± 0.631	16.8247 ^cfi^ ± 2.099	309 ^d^ ± 35.282	1.0877 ± 0.0002	7.2087 ^cdg^ ± 0.580

Note: Values are expressed as mean ± SD. ^a^ *p* < 0.05, ^b^ *p* < 0.01, ^c^ *p* < 0.001 vs. CTRL; ^d^ *p* < 0.05, ^e^ *p* < 0.01, ^f^ *p* < 0.001 vs. INF; ^g^ *p* < 0.05, ^h^ *p* < 0.01, ^i^ *p* < 0.001 vs. DICLO. Animal groups: CTRL—negative control, healthy rats; INF—positive control, rats with inflammation; PV groups—rats with inflammation treated with PV extract 100%, 50%, 25%, and 10%.

**Table 3 ijms-25-04843-t003:** Correlation coefficients between the studied redox status parameters.

	MDA–TOS	NO–TOS	Thiols–TAC	TOS–TAC	MDA–TAC	NO–TAC	Thiols–TOS
**CTRL**	−0.0830	0.9837 ^c^	0.6983	0.4854	−0.8416 ^a^	0.4642	0.9429 ^a^
**INF**	−0.0765	0.0623	0.9376 ^b^	0.2364	0.6591	0.7276	0.4228
**DICLO**	−0.1656	0.8552 ^a^	−0.0528	0.5661	−0.1964	0.1606	0.2124
**PV100**	0.3192	0.1857	0.0055	0.0901	0.3766	0.7177	0.4924
**PV50**	0.1544	0.9185 ^b^	−0.3895	0.3517	−0.3169	0.6434	0.5425
**PV25**	0.3777	0.0843	0.0000	0.26482	−0.6179	0.8827 ^a^	−0.8117 ^a^
**PV10**	0.5575	0.8404 ^a^	−0.2348	0.4747	0.4995	0.8624 ^a^	0.2599

Note: The values represent the correlation coefficient r/rho. ^a^ *p* < 0.05, ^b^ *p* < 0.01, ^c^ *p* < 0.001. Animal groups: CTRL—negative control, healthy rats; INF—positive control, rats with inflammation; PV groups—rats with inflammation treated with PV extract 100%, 50%, 25%, and 10%.

## Data Availability

Data are contained within the article.

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
