# Peer review of "Phytochemical Analysis and Antioxidant Effects of Prunella vulgaris in Experimental Acute Inflammation"

_ijms, 2024, doi:10.3390/ijms25094843_

Round 1
Reviewer 1 Report
Comments and Suggestions for Authors
Presented work is interesting for readers and give some valuable information about in vitro and in vivo activity of Prunella vulgaris extracts. However, the manuscript is really poorly written and should undergo systematic revision. English language is below any academic standard, meaning of some sentences is unclear and there are also some false claims. Here are some remarks:
Introduction
Line 41: There is no explanation what ROS stands for. Authors should correct it to reactive oxygen species (ROS).
Lines 43-46: English language should be improved in both sentences. There is no clear how measurements of specific biological biomarkers prevent damages and oxidative stress associated disorders? Measurements would not prevent anything.
Lines 46-48: "Malondialdehyde (MDA) that has a central role in oxidative stress and is used for assessing lipid peroxidation". This is false statement. MDA is just byproduct of lipid peroxidation and doesn't have central role at all.
Lines 49-52. English language should be improved in both sentences.
Lines 59-61: "Thus, inflammatory diseases biomarkers should involve not just inflammatory mediators, then also oxidation-related products and antioxidants". I don't have idea what this sentence mean. For example, "should involve not" have to be changed into "should not involve", and that is not the only grammar mistake in this sentence.
Lines 62-end of manuscript: English language used in this manuscript is incomprehensible.
Lines 62-64: "A World Health Organization (WHO) analysis found that about 80% of the world population use phytotherapy for their primary health care, and most of THESE use plant extracts and their active components."
Lines 67-68: "Prunella vulgaris L. [8]. P. vulgaris" instead of "Prunella vulgaris [8]. P. vulgaris L.". L. (for Linneaus) goes after the full name of species first time it appear in text, not after abbreviation.
Lines 76-77: "The infusion of wild basil is also used for the treatment of inflammations and lesions of the mouth and throat regardless of their cause". Wild basil?
Lines 79-81: The aim of this study is poorly written and it is not clear what authors wanted to achieve.
Results
Line 84: Ethanol extract instead of alcoholic extract.
Line 98: Why authors used only one antioxidant activity assay (DPPH)? It is well-known that two or three (at least) assays should be used to evaluate antioxidant activity of extracts.
Tables are not self-explanatory, readers could not know what all that abbreviations stands for.
Due to poor English language used it is hard to continue reviewing this manuscript. Almost every single sentence need to be improved. There are some false statments in the text that should be corrected.
Comments on the Quality of English LanguageIt is very difficult to read this mansucript. Manuscript should be rewritten completly and standard scientific English used.
Reviewer 2 Report
Comments and Suggestions for Authors
Please refer to the attached for review comments.

Considering readability, this manuscript requires significant grammatical improvement to enhance clarity and writing quality.
Reviewer 3 Report
Comments and Suggestions for Authors
IJMS Review
General Comments:
· Place all scientific names, including genus and species names, in italics
· Maintain consistency by using either 'PV' or 'P. vulgaris' throughout the manuscript. Review the text and ensure uniformity in the terminology chosen
· Commence a sentence with a word, not a number. For instance, instead of using '36,' write it out as 'thirty-six
Abstract:
· The format of the abstract is correct; however, the words Background, Methods, Results, and Conclusions should be omitted and the abstract should be formatted into a paragraph form.
· The DPPH test should be a DPPH radical antioxidant assay
· Please ensure that all instances of numbers with a comma as a decimal separator, such as '27,52 mM,' are corrected to use a period instead, as in '27.52 mM.' Review the manuscript and make the necessary adjustments.
· Maintain a space between units and numbers. Review the manuscript and make the necessary adjustments
· The result of the HPLC-DAD-ESI-MS should be included in the abstract
· In the results section of the abstract, indicate which concentration had shown the best and the lowest activity against the different tests performed.
· Improve the conclusion. Base your conclusion on the tests or analysis performed.
· Keywords:
Keywords:
· maximize the number of keywords to ensure better searchability and visibility of your paper. Include all the assays performed or the animal models used, the methods that you used like HPLC-DAD, ESI-MS, and the major group of compounds that you had identified, for example, phenolics, polyphenols, or flavonoids.
Introduction:
· The introduction is substantial but requires proper organization to maintain cohesiveness.
· The introduction can be subdivided into 3 or 5 paragraphs to enhance the readability and the clarity of thoughts.
· The introduction lacks sufficient justification for conducting total phenolic analysis and determining and quantifying phenolic compounds. Enhancements are necessary to address this gap
Results
· The results should be organized and presented chronologically based on how it was ordered in the methodology,
· Section 2.1. Show the chromatograms first before showing table 1.
· Figure 1 in Section 2.1. Show the chromatogram of the plant extract on the right side of the chromatogram of the standards. Do this for both UV-VIs wavelengths. Ensure that the x and y axis of the figures are properly labeled.
· In Table 1, the compounds are currently categorized into two classes: simple phenolics, which encompass hydroxybenzoic acids and derivatives, as well as hydroxycinnamic acids and derivatives, and flavonoids, including flavonols, flavones, flavanone, etc. To improve readability, it is recommended to introduce a new column classifying compounds into their respective classes. This additional classification will enhance organization and clarity for readers. Additionally, arranging compounds alphabetically within each subclass will streamline the presentation, facilitating easier reference for researchers. Lastly, labeling the last column as "Amount (μg/mL)" will specify the unit of measurement for the quantity of each compound, ensuring consistency and precision in reporting the concentration data.
· Section 2.1. The results showed here is very misleading. In your methodology, you indicated that you used gallic acid to express the amount of the total hydroxybenzoic acid derivatives, etc. However, your table shows that the compounds here have individual corresponding quantities. Improve the clarity of this as I have seen this as a huge flaw of your manuscript.
· Section 2.2. Improve the expression of results. What is the F value? You can indicate the Trolox equivalent and no need to indicate the percent inhibition.
· Section 2.3. Clarify the expression of results in this section. Rephrase for better readability and comprehension.
· No need to include the term statistically if you have already used the word significant
Discussion:
· In the first line of the discussion, you indicated that: The phytochemical analysis of the P. vulgaris ethanol extract found compounds with 150 antioxidant activity. How did you knew about this? Were you able to test the compounds for its antioxidant activity? If not, then you need to cite articles and add paragraphs that indicate the antioxidant activity of the identified compounds.
· Organize your discussion based on the chronological order of the methods and then the results.
·
Materials and Methods:
· I suggest that the authors needed to reorganize the order of the methodology. The phenolic compound determination should come last and the total phenolic content and DPPH Radical scavenging activity should come first.
· Section 4.1. It should be titled Chemicals and Reagents
· In Section 4.1, verify the spelling of chemical names and ensure that subscripted numbers are correctly applied to elements. Review the text meticulously for accuracy.
· Section 4.2. This section should only contain the information about the plant sample and should be in paragraph form. Also, revise clarify, and specify in detail the extraction method that was performed in the plant.
· Section 4.3.1. Total Phenolic Content Analysis is enough as a title
· Section 4.3.1. Revise the methodology for the TPC analysis. Start with the reagents used and then indicate how the final result was expressed.
· Make sure to indicate the instrument, supplier, and country of origin. Ensure that all the instruments used are documented in this manner.
· Section 4.3.1. How can you quantify if you use a wide range of UV-Vis scans? In your description of the Folin-Ciocalteu method, you indicated that the absorbance is read at 750 nm. This needs clarification.
· Section 4.3.2. Phenolic Compound Determination and Quantification is enough as a title.
· Section 4.3.2. Hydroxybenzoic acids were quantified as gallic acid equivalent, hydroxycinnamic acids as chlorogenic acid equivalent, flavanone as hesperidin equivalent, and flavonol and flavone as rutin equivalent. Specify the unit of gallic acid, etc. per amount of sample and the quantity of the sample used for quantification
· Section 4.3.2. The methodology did not explicitly indicate if how the different compounds were identified and quantified. Did you run the standards along with the unknown? Include this in the methods.
· Section 4.4. Citation is required for the statement regarding the mechanism of the DPPH assay.
· Section 4.4. convert the Trolox concentration to micromolar for better comprehension.
· Section 4.4. The working protocol should come after the description of the assay and should be in paragraph form.
· Section 4.4. and others. All formulas all throughout the manuscript should be formatted based on the MDPI format.
· n and p for sample number and p values should be italicized.
· Section 4.5.2. All the procedures must be written in detail to ensure the reproducibility of the experiment.
·
Conclusion:
· Improve the conclusion. Base your conclusion on the tests or analysis performed.
Comments on the Quality of English Language
I had embedded my comments regarding the English in my comments to the authors. I however needed to reiterate that the authors should read the paper properly and ensure that the statements are cohesive. Gammar and spelling errors should also be checked.
Reviewer 4 Report
Comments and Suggestions for Authors
The manuscript represents one more investigation of the chemical composition and biological activity of the ethanolic Prunella vulgaris extracts. As authors state themselves, on one hand, this plant has been known for ages to possess medicinal properties and, on the other hand, the content of the extract and concentration of the extracted compounds depend on the geographic location, plant harvesting time, part of the plant, etc. However, the reported results might be interesting for comparison reasons and may add to the global knowledge to support the ongoing interest in the natural sources for medical agents. Therefore, the manuscript could be considered for publication before improvement according to the suggestions as follows:
- Authors could add some reasoning for choosing concentration of 60% ethanol for extraction. Where the concentrations of the extracted compounds the highest at this concentration?
- Line 41: ROS needs full explanation at the first mentioning.
- Lines 157-158: it is wrong to say that “Gentisic acid is also known as 2,5-dihydroxybenzoic acid“ because 2,5-dihydroxybenzoic acid is a chemical name of this compound not just some another trivial name.
- Lines 313-331: the paragraph should be corrected for repeating compounds, their formulas (indexes in superscript), consistency in writing chemical formulas in the brackets after the chemical names in words; capital letters at the beginning of the word in the middle of the sentence; 2,2-Difenil-1-picrilhidrazil (DPPH) should be written correctly.
- Chapters 4.3.1, 4.3.2., and 4.3 need references.
Comments on the Quality of English LanguageThe English language must be revised. Some parts of the manuscript can't be read with pleasure. Here are some selected suggestions:
- Line 42: the correct word is “unstable” instead of “instable”
- Lines 45-46: grammar must be corrected, “can changes” is wrong combination.
- Line 60: “but” should be used instead of “then”
- Lines 79-80: the cluster of words “P. vulgaris ethanol extract phytochemical analysis,“ is too clumsy. Maybe at least it could be written as “phytochemical analysis of P. vulgaris ethanol extract“.
- Line 150: it is wrong to say “extract found compounds”.
- Lines 153-154: the sentence needs revision to be understandable easier.
- Lines 150-188 and elsewhere: the authors should be consistent in writing the chemical names of the compounds starting from the lower-case letter in the middle of the sentence.
- Lines 191-193: the sentence needs revision.
- Line 200: climate is the word to be used in a scientific text
- Lines 218-220: the sentence needs revision.
- Line 285: the repeating word “extract” should be deleted.
Round 2
Reviewer 2 Report
Comments and Suggestions for Authors
The manuscript received notable revisions from the authors and can be considered for publication. However, some minor issues still require the authors to make improvements.
L84, L110, L112, L241 and L307: Prunella vulgaris. In addition, there are others that use P. vulgaris. The authors have defined the abbreviation in L21, yet it is still inconsistent throughout the article. Please keep it consistent.
L431: "200nm" : Need a space.
L441-443, L448, and L458: Regarding the values range, again. Please use the half-stripe “–” Please correct this throughout the manuscript.
L507: "CONTROL" Please keep it consistent.
Reviewer 3 Report
Comments and Suggestions for Authors
IJMS Review
Abstract:
· The methods part of the abstract has to be revised into a better sentence format, particularly, the sentence containing the TPC, HPLC, and DPPH.
· The results section of the abstract has to be improved, particularly, on how the TPC result was presented and the result of the phenolic compound determination wherein the compounds have to be enumerated. Also, since you did not perform a test for the individual compounds, do not indicate that they have antioxidant properties.
· The units of the TPC result should be consistent with the results that you had shown in the results section. Also, spell out GAE and enclose the abbreviation in a parenthesis,
· In the results section of the abstract, indicate which concentration had shown the best and the lowest activity against the different tests performed.
Introduction:
· Line 58, Page 2. Spell out SOD, CAT, and GPx and enclose its respective abbreviations in a parenthesis.
· Combine 2nd and 3rd paragraphs, lines 77-90 and 95 to 99.
· Separate lines 90 to 94 from the previous placements and add a summary of compounds that were reported in the plant and relate it to lines 90 to 94.
Results
· Line 111, Page 3. 0.28 mg GAE/1mL extract should be 0.28 mg GAE/mL.
· Figure 1: Label the compounds in the chromatogram. you can include a see Table 1 for the identity of compounds and just label the peaks with a number or you can label the peaks directly.
· Also, the current format of the figure is not clear. I suggest place labels A-D for each figure and then describing the labels A-D in the description/title of the figure
· Dihydroxybenzoic acid, protocatechuic acid, vanillic acid, and rosmarinic acid are also simple phenolic acids.
· Line 117, page 5. statistically significantly. No need to include the term statistically if you have already used the word significant. PLEASE TAKE THIS comment very seriously.
· What is INF? Spell out on the first mention.
· Line 145, Page nM/L is a wrong unit. Fix this.
· Why were the individual compounds not quantified directly, and why were the results expressed in terms of gallic acid equivalents, chlorogenic acid equivalents, etc.
Discussion:
· Compare the compounds identified in PV with those found by other researchers.
Comments on the Quality of English Language
The authors should conduct thorough proofreading of their work to identify and correct errors in spelling and grammar. Additionally, they should focus on enhancing the cohesion of their statements and improving the overall readability of the text
Round 3
Reviewer 3 Report
Comments and Suggestions for Authors
IJMS Review_2945906_V3
Abstract:
· Indicate the unit for gallic acid. Is it mg? microgram?
· When you say individual representatives, does this mean specific? Pleae clarify
Results:
· In table 2 and all thorugh out the manuscript, please take note that µM/L is the wrong unit!!!! M is already a unit derive from molar mass over volume!
Comments on the Quality of English LanguageDear Editors,
I have very few comments on this latest revision, however, I do hope that you will make sure that the unit µM/L will be changed accordingly because this unit is wrong.
Thank you and kind regards,
Ivan
